New information on the early Permian lanthanosuchoid Feeserpeton oklahomensis based on computed tomography

MacDougall Mark J. 1 mark.macdougall@mfn.berlin
Winge Anika 1
Ponstein Jasper 1
http://orcid.org/0000-0002-9211-9361 Jansen Maren 1 2
http://orcid.org/0000-0002-7454-1649 Reisz Robert R. 3
Fröbisch Jörg 1 4
1 Museum für Naturkunde Leibniz-Institute für Evolutions- und Biodiversitätsforschung , Berlin , Germany
2 Museum für Naturkunde Magdeburg , Magdeburg , Germany
3 University of Toronto Mississauga , Mississauga, ON , Canada
4 Humboldt-Universität zu Berlin , Berlin , Germany
Hutchinson John
Electronic publication date: 2019 Oct 31
Publication date: 2019
Volume: 7
Electronic Location ID: e7753
Received 2019 May 16; Accepted 2019 Aug 26
Copyright: © 2019 MacDougall et al.
Copyright year: 2019
Copyright holder: MacDougall et al.
License: This is an open access article distributed under the terms of the Creative Commons Attribution License, which permits unrestricted use, distribution, reproduction and adaptation in any medium and for any purpose provided that it is properly attributed. For attribution, the original author(s), title, publication source (PeerJ) and either DOI or URL of the article must be cited.
License URL: https://creativecommons.org/licenses/by/4.0/

Keywords: Reptilia, Cisuralian, Parareptilia, Palaeozoic, Sauropsida

Funding: Leibniz-DAAD postdoctoral scholarship and currently a Humboldt postdoctoral fellowship Research Track Scholarship of the Humboldt-Universität zu Berlin within the Excellence initiative of the states and the federal government National Science and Engineering Research Council (NSERC) Discovery grant German Research Foundation (DFG) Mark J. MacDougall was supported by a Leibniz-DAAD postdoctoral scholarship and currently a Humboldt postdoctoral fellowship, Jasper Ponstein is supported by a Research Track Scholarship of the Humboldt-Universität zu Berlin within the Excellence initiative of the states and the federal government, Robert R. Reisz is supported by a National Science and Engineering Research Council (NSERC) Discovery grant, and Jörg Fröbsich is supported by the German Research Foundation (DFG). The funders had no role in study design, data collection and analysis, decision to publish, or preparation of the manuscript.

==============================
The cave deposits of the Lower Permian Richards Spur locality in Oklahoma, USA, have produced an incredible number of terrestrial tetrapod taxa, many of which are currently only known from this locality. One of the many recent taxa to be described from the locality was the small lanthanosuchoid parareptile Feeserpeton oklahomensis. Represented by a well-preserved, near complete skull, F. oklahomensis would have been a small predatory reptile, likely preying upon arthropods, and contributes to the extensive tetrapod fauna that was present at Richards Spur. New computed tomography data of the holotype and only specimen has allowed us to visualize and describe previously obscured and inaccessible anatomy of this taxon. These areas include the mandibular ramus, the palate, the sphenethmoid, the epipterygoids, and the braincase. Furthermore, this new anatomical information allowed formerly unknown character codings to be updated, thus we also performed new phylogenetic analyses that incorporated this new information. The results of these updated phylogenetic analyses are very similar to those of past studies, with F. oklahomensis being found as the sister taxon to all other lanthanosuchoids.

Introduction

The Lower Permian (Cisuralian) Richards Spur locality of southwestern Oklahoma, represented by an extensive cave system, is known for its immense terrestrial tetrapod fauna (Sullivan & Reisz, 1999; MacDougall et al., 2017b). Over the last few decades, more than 30 taxa have been described from the locality (MacDougall et al., 2017b), which includes various anamniote tetrapods, synapsids, and reptiles. Among these tetrapods are numerous parareptile taxa, specifically: Colobomycter pholeter Vaughn, 1958, Delorhynchus priscus Fox, 1962, Bolosaurus grandis Reisz, Barkas & Scott, 2002, Microleter mckinzieorum Tsuji et al., 2010, Abyssomedon williamsi MacDougall & Reisz, 2014, D. cifelli Reisz et al., 2014, and C. vaughni MacDougall et al., 2016. Most early Permian continental assemblages exhibit only a single parareptile taxon; Richards Spur is in stark contrast to this pattern, with eight parareptiles being known from the locality (MacDougall et al., 2017b), several of which belong to the clade Lanthanosuchoidea (Vaughn, 1958; Fox, 1962; Modesto, 1999; MacDougall & Reisz, 2012; Reisz, MacDougall & Modesto, 2014; MacDougall, Modesto & Reisz, 2016).

Lanthanosuchoidea is a small clade of reptiles that includes various early and middle Permian forms. Currently, all early Permian lanthanosuchoids are only known from North America, whereas as the middle Permian taxa are known only from Russia. There are also notable differences between the North American and Russian lanthanosuchoids. The early Permian North American taxa are considered to be small, fully-terrestrial, predatory taxa (Modesto, Scott & Reisz, 2009; Haridy, MacDougall & Reisz, 2017; MacDougall et al., 2017a), whereas the middle Permian Russian taxa are larger and considered to be semi-aquatic (Sennikov, 1996; Reisz, 1997; Verrière, Brocklehurst & Fröbisch, 2016). Overall, this suggests that the clade could have potentially originated in western Laurasia and later dispersed to eastern Laurasia, though the lack of Russian early Permian strata makes this biogeographic hypothesis difficult to test.

The lanthanosuchoid Feeserpeton oklahomensis is one of the most recently described taxa from Richards Spur locality and is currently only known from its holotype (MacDougall & Reisz, 2012). It is represented by a small, nearly complete skull, and is characterized by several enlarged maxillary teeth on the maxillae and dentaries, as well as large postorbitals and small squamosals. In the initial description, MacDougall & Reisz (2012) largely described the visible external anatomy of the skull, although computed tomography (CT) data was used to examine the mandibular dentition. However, they did not examine any other areas of the skull that were obscured or inaccessible, which left some aspects of the anatomy of the skull unknown.

Herein, we describe the previously inaccessible anatomy of F. oklahomensis using new CT data. Areas that were segmented and examined include obscured parts of the lower jaw and palate, the sphenethmoid, the epipterygoids, and elements of the braincase (Fig. 1). Furthermore, the new information obtained from this data also warranted updating the phylogenetic character codings of F. oklahomensis, thus updated phylogenetic analyses were performed as well.

Figure 1 The skull of Feeserpeton oklahomensis, OMNH 73541, showing the regions that were reconstructed using CT data.

Scale bar equals two mm.

Materials and Methods

Specimen

The skull of F. oklahomensis examined for this study is the holotype and currently only known specimen, OMNH 73541. It was previously studied and described by MacDougall & Reisz (2012).

Computed tomography data

The skull of F. oklahomensis was scanned using the X-ray CT setup (Phoenix ǀ X-ray Nanotom ǀ s) at the Museum für Naturkunde in Berlin. Scan parameters were set to 57 kV voltage and 170 µA current with 1,440 images/360° at an exposure time of 1,000 ms/image and an effective voxel size of 0.0129 mm, resulting in a magnification rate of 3.857. Cone beam reconstruction was performed using datos ǀ X-ray sensing four Inspection Technologies GmbH (phoenix ǀ X-ray) with a correction value of 1.845. The elements were visualized and digitally segmented in VG studio Max 3.2. Smoothing of elements was accomplished by rendering the images using the volume rendering setting “Phong” with an oversampling of 2.

Phylogenetic analyses

The data matrix used in the two phylogenetic analyses is based on the data matrix of MacDougall et al. (2018). The analyses were performed in PAUP 4.0a165 (Swofford, 2003), using parsimony as the optimality criterion. The outgroup was set to include the taxa Seymouria, Limnoscelis, and Orobates. Furthermore, minimum branch lengths of zero were set to collapse. For the first analysis a heuristic search with tree bisection and reconnection (TBR) branch swapping was performed, with the addition sequence algorithm set to simple, and multistate characters unordered. Both a bootstrap (1,000 replicates) and Bremer decay analysis were conducted to determine support values for clades.

We also performed a second analysis using the methodology documented by Laurin & Piñeiro (2018) to see how the ordering of characters would affect tree topology and support values.

The character codings of F. oklahomensis were updated based on new information obtained from the CT data. The characters that were recoded are as follows: interpterygoid vacuity anterior extent (61) ?⇾0, alar flange of the vomer presence or absence (64) ?⇾0, cultriform process present; (67) ?⇾1, pterygoid anterior extent (85) ?⇾0, cultriform process anterior extent (86) ?⇾1, sutural contact between paroccipital process and dermatocranium (92) ?⇾1, Meckelian fossa orientation (108) ?⇾0, Meckelian fossa anteroposterior length (109) ?⇾0, number of coronoids (113) ?⇾1, prearticular anterior extent (114) 1⇾0, coronoid process height (119) ?⇾0, high coronoid process composition (120) ?⇾-, single large tooth on anteriormost end of vomer (169) 1⇾0. Furthermore, a few characters were recoded for other taxa to correct some coding errors; these changes are indicated in the data matrix. The updated data matrix and the full character list can be found in the Supplemental Information.

Description

The CT data of F. oklahomensis allowed for several previously inaccessible and obscured areas of the skull to be examined and segmented (Fig. 1). These areas include parts of the palate and the mandible, the sphenethmoid, the epipterygoids, and elements of the braincase.

The palate

Many of the palatal elements of F. oklahomensis are visible in the holotype; however, only the right side of the palate was prepared, and there are portions of it that are obscured by supportive matrix. Through segmentation of the CT data both sides of the palate are visible in their entirety (Fig. 2), some parts of the palate are damaged or not preserved, but it is largely intact. The most notable feature of the palate that is clarified is the extent of the palatal dentition, in particular on the palatine. The left palatine appears to be largely complete and grants us a view of the element that was not available previously.

Figure 2 The palate of Feeserpeton oklahomensis, OMNH 73541, reconstructed from CT data.

(A) Dorsal view, and (B) ventral view. Abbreviations: ec, ectopterygoid; m, maxilla; pal, palatine; ps, parasphenoid; pt, pterygoid; v, vomer. Scale bar equals four mm.

As was suggested by MacDougall & Reisz (2012), the palatine is a large element and makes up a considerable portion of the overall palatal surface. Anteriorly, the palatine contacts the vomer, whereas it contacts the pterygoid medially and posteriorly, the ectopterygoid is also contacted posteriorly. Laterally, there is a contact with the maxilla. MacDougall & Reisz (2012) note that there are two clusters of palatine teeth visible, though the full extent of these clusters could not be fully determined. The segmented left palatine clearly shows that there is a lateral and medial group of teeth on the element; they are roughly organized in the form of rows. The lateral row consists of seven large teeth and a few smaller ones, whereas the medial row consists of several smaller teeth.

The full anteroposterior extent of the vomer is also now apparent, the element is about the same length as the palatine, and ends anteriorly with a narrow, pointed medially positioned process. Likewise, the full lateral extent of the element reveals that the anterior end of the palatine is nestled between two posterior extensions of the vomer: a broad medial extension, and a narrower lateral one.

There are other features of the palate that are clarified with this new CT data. It is now clear that the interpterygoid vacuity extends quite far anteriorly, past the posterior edge of the palatines. Likewise, it is apparent that the cultriform process of the parasphenoid extends forward for much of the length of the vacuity. Furthermore, the dorsal surface of the parasphenoid is now visible for the first time, revealing several previously unknown aspects of its anatomy. Anteriorly, where the cultriform process merges with the body of the parasphenoid, a small pit, the sella turcica, is visible. Immediately posterior to this pit is the dorsum sellae, a slightly raised wall that borders the sella turcica. Unfortunately, due to the poorer resolution of the CT data in this area, certain features of the parasphenoid were not able to be segmented, specifically the small teeth that are present on this element (MacDougall & Reisz, 2012). Furthermore, a suborbital foramen is not present, with the palate being unbroken at the intersection of the pterygoid, palatine, and ectopterygoid, as is the case in other lanthanosuchoids (Debraga & Reisz, 1996; Reisz, MacDougall & Modesto, 2014).

The mandible

Due to the mandibular rami of F. oklahomensis being occluded with the upper jaw, only details of the ventral and labial surfaces can be clearly observed, with much of the dorsal and lingual surfaces being obscured by the occlusion. However, segmentation of the entire left mandibular ramus (Fig. 3) reveals new information regarding these previously inaccessible areas. The left ramus is slightly damaged on its labial surface, which results in a small gap through which the labial surface of the prearticular can be seen.

Figure 3 The left mandibular ramus of Feeserpeton oklahomensis, OMNH 73541, reconstructed from CT data.

(A) Dorsal view, (B) ventral view, (C) labial view, and (D) lingual view. Abbreviations: an, angular; ar, articular; c, coronoid; d, dentary; pra, prearticular; sa, surangular; sp, splenial. Scale bar equals two mm.

MacDougall & Reisz (2012) used CT data to investigate the dentition of the dentary, though only the shape of the teeth was examined. The segmented mandibular ramus of the holotype exhibits 21 teeth on the dentary, however, when empty alveoli are included the total number of dentary teeth increases to 25. Furthermore, there are enlarged teeth found on the anterior end of the element; there are two visible and an empty alveolus, suggesting that there would have been three enlarged teeth, the same pattern that is observed on the maxilla (MacDougall & Reisz, 2012).

One of the more notable features of the mandible are the coronoid elements (Figs. 3 and 4). In their investigation of multiple coronoids and coronoid dentition in Palaeozoic reptiles, the coronoid of F. oklahomensis was examined by Haridy, MacDougall & Reisz (2017) using CT data. They identified the presence of two coronoids on each ramus, as well as the presence of coronoid dentition. We are able to confirm that coronoid dentition is present in F. oklahomensis (Fig. 4), though much of the dentition appears to not be preserved, there are three small teeth that are clearly visible on the right coronoid. However, unlike what was observed by Haridy, MacDougall & Reisz (2017), we find no trace of two coronoid elements (Fig. 3), suggesting that the presence of multiple coronoids in F. oklahomensis was likely a misinterpretation.

Figure 4 The right coronoid of Feeserpeton oklahomensis, OMNH 73541, reconstructed from CT data.

(A) Lingual view, and (B) dorsal view. Scale bar equals one mm.

The coronoid itself is a relatively long element found lingual to the posterior end of the dentary. It extends anteriorly from its posterior articulation with the surangular to the 18th tooth position of the dentary, narrowing for most of its length. A posteroventral process of the coronoid curves posteriorly to meet with the prearticular.

The splenial is largely restricted to the medial surface of the mandibular ramus (Fig. 3), making it difficult to examine fully previously. Anteriorly, the two splenials would have met with another and contributed to the mandibular symphysis. The anterior end of the splenial also exhibits the presence of the foramen intermandibularis oralis. Usually, in early reptiles there was also a foramen intermandibularis caudalis found at the intersection of the splenial, angular, and prearticular, but it does not appear to be present in this specimen of F. oklahomensis, which may be the result of the slight damage present in this area.

The full extent of the prearticular, an element not described by MacDougall & Reisz (2012), is revealed in the segmented mandibular ramus (Fig. 3). It is a long element that extends from the articular to the posterior end of the splenial. It also articulates ventrally with the portion of the angular that wraps around to the medial side of the ramus, whereas its anterodorsal edge contacts the coronoid and dentary. The dorsal portion of the prearticular forms the medial margin of the large adductor fossa. Overall, the prearticular is quite similar in shape and position to that of other closely related taxa, such as Delorhynchus (Haridy, MacDougall & Reisz, 2017).

The Meckelian fossa, normally completely obscured by the occlusion of the jaws to the rest of the skull, is revealed in its entirety in the segmented mandibular ramus as well (Fig. 3). The Meckelian fossa of F. oklahomensis is found on the posterior end of the mandibular ramus. The fossa faces dorsomedially and it is quite long, extending anteriorly for about a third of the length of the ramus, as is the case in Delorhynchus (Haridy, MacDougall & Reisz, 2017).

The sphenethmoid

The sphenethmoid is a rarely described element in early reptiles, usually because it is either not preserved or not visible. Due to the internal position of the sphenethmoid in the holotype of Feeserpeton it was not noted or discussed in MacDougall & Reisz (2012), here we present the fully segmented sphenethmoid element (Fig. 5).

Figure 5 The sphenethmoid of Feeserpeton oklahomensis, OMNH 7354, reconstructed from CT data.

(A) Anterior view, (B) posterior view, (C) dorsal view, and (D) right lateral view. Scale bar equals one mm.

The sphenethmoid is found in its expected position, ventral to the frontals. It is a roughly Y-shaped element when viewed in anterior and posterior aspects, possessing a slender ventral process (often termed the keel) and two equally slender dorsal processes with a rounded trough between them. Lateral view of the element reveals that its anteroposterior length is about equivalent to its dorsoventral height. This sphenethmoid shape is similar to what has been observed in other early reptiles, such as Captorhinus (Heaton, 1979; Modesto & Reisz, 2008).

The epipterygoids

The epipterygoids are another example of rarely described elements of early reptiles, largely due to their interior position within the skull. The new CT data reveal the presence of both epipterygoids in the holotype of F. oklahomensis, which were both fully segmented for the purpose of this study (Fig. 6). These elements were not described by MacDougall & Reisz (2012) as they are not exposed externally. Both epipterygoids appear to be disarticulated and not in their natural positions (Fig. 1), but overall they are similar in structure to those that have been described for other early reptiles (Romer, 1956; Carroll & Lindsay, 1985), possessing a gracile dorsal columella that arches slightly posterior, and a ventral region that expands to form a more robust footplate with a broad anteroposterior length. The footplate also exhibits a process on its anteromedial surface, a characteristic that has also been observed in other early reptiles, such as Captorhinus (Fox & Bowman, 1966). In their natural positions the dorsal columella of each epipterygoid would have contacted the supraoccipital and prootic, whereas the footplate would have presumably met with the quadrate process of the pterygoid, as is the case in Captorhinus (Fox & Bowman, 1966) and the procolophonids Leptopleuron and Hypsognathus (Sues et al., 2000; Cisneros, 2008).

Figure 6 The left epipterygoid of Feeserpeton oklahomensis, OMNH 73541, reconstructed from CT data.

(A) Lateral view, (B) medial view, (C) dorsal view, and (D) ventral view. Scale bar equals one mm.

The braincase

The braincase of F. oklahomensis was described externally by MacDougall & Reisz (2012), however, there was still substantial portions of its overall anatomy that could not be described due to being obscured. The fully segmented braincase presented here (Fig. 7) reveals several details that could not be observed in the original description of the holotype. It is also quite apparent that the braincase exhibits slight disarticulation of elements from their natural position, as well as some damaged areas.

Figure 7 The braincase of Feeserpeton oklahomensis, OMNH 73541, reconstructed from CT data.

(A) Dorsal view, (B) ventral view, (C) right lateral view, and (D) posterior view. Abbreviations: bo, basioccipital; eo, exoccipitals; op, opisthotic; pro, prootic; ps, parasphenoid; so, supraoccipitalst, stapes; ?, unknown fragment that may be part of the left stapes or the left opisthotic. Scale bar equals two mm.

The prootics of Feeserpeton are large, irregularly shaped anterior elements of the braincase and form an extensive portion of its overall structure. While MacDougall & Reisz (2012) were able to identify the prootics, they were only able to describe the exposed ventral surface of the elements, which is only a small portion of the overall element. The prootics appear to be entirely intact but are slightly disarticulated and not in contact with most of the other elements of the braincase, though laterally they do have a slight contact with their respective opisthotics. The posterior surface of the prootic would have articulated with the anterior surface of the opisthotic. The flattened posterolateral extension of the element, usually termed the paroccipital process of the prootic would have met with the paroccipital process of the opisthotic, with a large depression in this region of the prootic contributing to the fenestra ovalis. Dorsomedially, there is a small process that would have met with the supraoccipital. The medial portion of the element exhibits another smaller process that extends ventrally out from the main body of the prootic, which would likely have come close to contacting the basioccipital. The anterior part of the prootic is convex in shape. Overall, the structure of the prootic is largely similar to what has been described for other parareptiles, such as Leptopleuron (Spencer, 2000), though in the case of Leptopleuron the prootic appears to be not quite as robust as that of Feeserpeton. Unfortunately, this is a rarely described element of closely related taxa, due to its often-inaccessible position.

The opisthotics are another large component of the braincase, and like the prootics, they were also only partially described by MacDougall & Reisz (2012). The segmented braincase clearly illustrates that both of the opisthotics suffer from damage (Fig. 7); however, the left opisthotic is substantially more damaged than the right one, with a large posterior segment of the element being completely absent. The better preserved right opisthotic clearly exhibits a laterally expanded anterodorsal end, and a posterior portion that extends ventrally. The broad anterior end contributes to the paroccipital process of the opisthotic, which would have met with the paroccipital process of the prootics. Moving medially the posterior portion of the opisthotic expands dorsoventrally and meets with the lateral edge of the fused exoccipital-basioccipital complex, contributing to the posterior end of the braincase and forming the remaining portion of the paroccipital process.

The stapes was briefly described by MacDougall & Reisz (2012), and we expand upon their description here. The stapes of Feeserpeton is not in its natural position, being slightly disarticulated; it consists of a bifurcating laterally compressed distal end, and the broad proximally located footplate, which are connected by a short, slightly twisted shaft. In its natural position the footplate would have met with the fenestra ovalis of the paraoccipital process, with the distal end extending laterally towards the quadrate. The distal end of the stapes bifurcates into two distinct processes, the distal facing columella, which appears to be broken, and prior to this process a dorsal extension. The stapes is similar in size and structure to the stapes of Acleistorhinus pteroticus (Debraga & Reisz, 1996), but is quite distinct from the small gracile stapes of Leptopleuron lacertinum (Spencer, 2000).

Phylogenetic analyses

The first phylogenetic analysis produced 27 optimal trees, each with a tree length of 667. As in the original MacDougall & Reisz (2012) study, the strict consensus tree (Fig. 8) produced from the optimal trees has F. oklahomensis being recovered as the sister taxon of all other lanthanosuchoids (it is in this position in all 27 of the optimal trees). The clade Lanthanosuchoidea is recovered as the sister taxon to the clade that contains Bolosauria, Procolophonoidea, Pareiasauridae, Nycteroleteridae, Nyctiphruretidae, and Microleter. This is similar to what was recovered in the analysis of MacDougall & Reisz (2012), except for the inclusion of Microleter within this clade. However, the position of Bolosauria in our analysis differs from what was recovered by other studies (Modesto et al., 2015; MacDougall et al., 2017a), where the clade was found to be more basal than Lanthanosuchoidea.

Figure 8 Strict consensus tree obtained from the phylogenetic analysis.

Tree length = 667, consistency index = 0.301, rescaled consistency index = 0.195, retention index = 0.647. Nodes of clades of interest are labeled: (A) Amniota; (B) Reptilia; (C) Parareptilia; (D) Eureptilia; (E) Lanthanosuchoidea; (F) Bolosauria; (G) Nyctiphruretidae; (H) Nycteroleteridae; (I) Pareiasauridae; (J) Procolophonoidea. Bootstrap support values are found above nodes, if no value is indicated it was less than 50%. Bremer support values are found below nodes, if no value is indicated the clade collapsed with the addition of one extra step.

Similar to the results of MacDougall et al. (2018), the ophiacodontid (Archaeothyris) and varanopid taxa (Archaeovenator, Mycterosaurus) included in the analysis were recovered as being more closely related to the reptile taxa than to the synapsids. This is an atypical result that warrants further investigation in the future, but is at least partly in line with the results of recent work that has recovered varanopids within Reptilia (Ford & Benson, 2018). In the case of our study this result could potentially be the result of the relatively narrow focus of our analysis.

The second analysis in which characters were ordered following the methodology of Laurin & Piñeiro (2018) produced nine optimal trees, the consensus of which resulted in a very similar tree (Fig. S1) to that recovered in our first analysis. The only difference being the resolution of the polytomy that contained the nyctiphruretids, procolophonoids, pareiasaurs, and nycteroleters.

Discussion

New information from CT data

The early Permian Richards Spur locality has produced a considerable amount of well-preserved fossil material, largely due to the unique preservational environment associated with the caves found there (MacDougall et al., 2017b). In particular, near complete skulls are not uncommon at the locality, with many taxa being known almost solely from skulls (Modesto & Reisz, 2008; Anderson, Scott & Reisz, 2009; Tsuji, Müller & Reisz, 2010; Polley & Reisz, 2011; MacDougall et al., 2017b). These various well-preserved specimens provide substantial information regarding the anatomy of the taxa to which they belong, however, there are also various regions of these skulls that cannot be examined normally, either due to being obscured or not being exposed externally. CT data are proving to be an ideal way to study these skulls and their difficult to examine areas, which will in turn provide more information regarding the Richards Spur assemblage and the taxa that compose it. The new CT data of the holotype of F. oklahomensis have allowed for the segmentation of several previously inaccessible areas, revealing more details regarding the anatomy of this taxon. Specifically, the dentition of the mandible and palate, and various elements of the skull that are largely internal.

Recently, there have been other parareptile taxa that have been examined using CT data (Tsuji, Sobral & Müller, 2013; Zaher, Coram & Benton, 2018), which resulted in the discovery of new information about the examined taxa, however, there are still numerous parareptile taxa that have yet to be examined in comparable detail using CT scanning. The new information that will be obtained from examining other parareptiles using CT scans will be important for resolving issues and testing existing hypotheses regarding early amniote relationships (Laurin & Piñeiro, 2017, 2018; Ford & Benson, 2018; MacDougall et al., 2018), as well as for better understanding the anatomy and evolution of these taxa (Zaher, Coram & Benton, 2018).

The coronoid eminence of Feeserpeton

One of the aspects of the anatomy of F. oklahomensis that was clarified with CT data is the composition of the coronoid eminence. The coronoid eminence of lanthanosuchoids has been shown to be quite complex, with the presence of multiple coronoids and dentition present on the elements (Haridy, MacDougall & Reisz, 2017; MacDougall et al., 2017a). Our identification of a single coronoid on the mandibular ramus of F. oklahomensis (Fig. 3), which is contrary to the observations of Haridy, MacDougall & Reisz (2017), does not dramatically change their interpretations about the evolution of the trait within Lanthanosuchoidea. The only potential change this reinterpretation introduces to their hypothesis is that the presence of multiple coronoids does not appear to be primitive for Lanthanosuchoidea, instead appearing later in the evolution of the clade. However, we were able to confirm that the coronoid of F. oklahomensis does indeed exhibit dentition (Fig. 4), thus it is clear that denticulate coronoids are the primitive condition for lanthanosuchoids, as was hypothesized by Haridy, MacDougall & Reisz (2017). Other lanthanosuchoid taxa, notably Acleistorhinus pteroticus, will have to be reexamined in the future to further elucidate details regarding the evolution of multiple and denticulate coronoids within the clade.

Conclusions

Through CT data and modern visualization techniques this study reveals new information regarding the anatomy of the early Permian reptile F. oklahomensis. Notably, we were able to describe numerous details regarding the anatomy of the mandibular rami, the palate, the sphenethmoid, the epipterygoids, and the braincase. All of which are parts of the skull that cannot be fully examined normally in the holotype specimen. This new information also allowed for several previously unknown phylogenetic characters to be coded. The evolution and relationships of early amniotes is still an area of paleontological research that is far from set in stone, and the investigation of normally inaccessible anatomy using CT data has been and will continue to be important for better understanding reptile, and more broadly, early tetrapod evolution.

Supplemental Information

Supplemental Information 1 Strict consensus tree obtained from the phylogenetic analysis with multistate characters ordered.

Tree length = 674, consistency index = 0.298, rescaled consistency index = 0.193, retention index = 0.646. Bootstrap support values are found above nodes, if no value is indicated it was less than 50%. Bremer support values are found below nodes, if no value is indicated the clade collapsed with the addition of one extra step.

Click here for additional data file.

Supplemental Information 2 Data matrix used in the phylogenetic analysis.

Click here for additional data file.

Supplemental Information 3 List of characters used in the phylogenetic analysis.

Click here for additional data file.

We would like to thank the Richard Cifelli and William May, for their assistance in acquiring this and other specimens, we also thank Mike Feese for his assistance, and his donating of many Richards Spur specimens to the OMNH. We also thank Sean Modesto, Michel Laurin, and Juan Cisneros for reviewing the manuscript and providing very helpful comments.

Institutional Abbreviations

OMNH Sam Noble Oklahoma Museum of Natural History, Norman, Oklahoma, USA.

Additional Information and Declarations

Competing Interests

Author Contributions

Data Availability

The authors declare that they have no competing interests.

Mark J. MacDougall conceived and designed the experiments, performed the experiments, analyzed the data, prepared figures and/or tables, authored or reviewed drafts of the paper, approved the final draft.

Anika Winge performed the experiments, analyzed the data, prepared figures and/or tables, approved the final draft.

Jasper Ponstein performed the experiments, analyzed the data, prepared figures and/or tables, approved the final draft.

Maren Jansen performed the experiments, analyzed the data, approved the final draft.

Robert R. Reisz conceived and designed the experiments, contributed reagents/materials/analysis tools, approved the final draft.

Jörg Fröbisch conceived and designed the experiments, contributed reagents/materials/analysis tools, approved the final draft.

The following information was supplied regarding data availability:

The data matrix and character list used in the phylogenetic analysis are available in the Supplemental Files.

Raw CT scans are available at MorphoSource: OMNH:73541, Feeserpeton oklahomensis, http://www.morphosource.org/Detail/MediaDetail/Show/media_id/42981.

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
