# Peer review of "New information on the early Permian lanthanosuchoid Feeserpeton oklahomensis based on computed tomography"

_PeerJ, doi:10.7717/peerj.7753_

## Round 0.1 · original submission · Major Revisions

Apologies for a bit of delay to completing the review process, but we did get 3 reviews for this nice paper, and they offer some constructive critiques. One also challenges the citation of some relevant work regarding ordering of multi-state characters, which could influence the findings. Please address all points individually in your Rebuttal. We look forward to receiving your revised MS.

·

Basic reporting

Acceptable.

Experimental design

Acceptable.

Validity of the findings

Mostly acceptable. Anatomical description should be augmented (see Comments for the author).

Additional comments

All in all, this is a good descriptive and phylogenetic update, using CT data to glean information from physically inaccessible regions of a type skull, and it will be a very useful and well cited contribution to the paleontology of early reptiles. With some corrections and additions to the description, this submission will make a useful and well cited publication in PeerJ.

What is missing: the CT images are fine but a few elements received little or no attention. For instance, the dorsal surface of the parabasisphenoid is finally revealed but there is no description of major landmarks such as the dorsum sellae, cristae trabeculares, etc. Surprisingly, the parasphenoidal teeth seen in figure 1 of MacDougall and Reisz (2012) are not shown/segmented out at all.

Finally, and most interestingly, the authors use the term ‘sauropsid’ for parareptiles, diapsids, et al. throughout their work, and the term ‘reptile’ appears only in the bibliographic entries (and, outside of the bibliography, the name ‘Reptilia’ appears only as a key word). This is in contrast to MacDougall et al. (2018), who preferentially used the term ‘reptile’ and eschewed the term ‘sauropsid’ for members of this synapsid sister group. Therefore, I find the present authors extensive use of ‘sauropsid’ a little odd, for the following reasons.

Although the authors do not state which concept (phylogenetic definition) of Sauropsida that they use, I surmise that they are using the Laurin and Reisz (1995) definition (‘Sauropsida is the last common ancestor of mesosaurs, testudines, and diapsids’) because they do not include Archaeothyris, Archaeovenator and Mycterosaurus (i.e. the three varanopid taxa) as sauropsids in ‘their’ sauropsid clade. However, Gauthier (1994) had previously defined Sauropsida as “reptiles plus all other amniotes more closely related to them than they are to mammals” (inasmuch Gauthier espoused a crown-group concept/definition for Reptilia). Gauthier’s definition tried to encapsulate Huxley’s concept for Sauropsida but Laurin and Reisz (1995), by including mesosaurs in their definition of Sauropsida simply made a hash of things, especially in light of the ongoing phylogenetic lability of turtles (and current Phylocode recommendations). I think the authors need to explain why they are using a phylogenetic definition for Sauropsida which does not have priority (otherwise they are simply cherry picking clade names here). Oh, and please clarify what a reptile is in their study.

More detailed comments/corrections:

165–169: if these two rows are (or are not) the palatine portion of the pterygo-palatine tooth cluster, this should be stated clearly.

178–180: I am not entirely convinced that a suborbital foramen is absent. The authors note that the ‘intersection of the pterygoid, palatine, and ectopterygoid is unbroken’, yet the pterygoid does not contribute to the suborbital foramen, which in early reptiles is usually formed by the palatine and the ectopterygoid (e.g. Coletta) or the palatine and the jugal (in most captorhinids, in which the ectopterygoid is fused to the jugal). In Coletta, the suborbital foramen is tiny, and as there is not a good fit between the palatines and the ectopterygoids in the Feeserpeton scan I am not sure there is compelling evidence to state that the suborbital foramen is genuinely absent.

183: each gnathostome has a single mandible (divisible into two mandibular rami); also, description of the mandible should follow that of the skull (palate, braincase, etc.).

184–222: the splenial morphology is not mentioned at all here. Is there evidence for a foramen intermandibularis oralis? Do the splenial, the prearticular, and the angular for a foramen intermandibularis caudalis, as they do in Delorhynchus cifellii and other early reptiles?

189: use ‘lateral’ rather than ‘labial’ for describing this surface of the prearticular (a ‘post-dentary’ bone)

191: clumsy use of the word ‘however’ here

207: ‘The coronoid’ . . . ‘extends from its posterior articulation with the surangular to the middle of the dentary’. Please specify as far anteriorly as to which dentary tooth position.

232: is this a left or a right lateral view of the sphenethmoid in figure 5?

237–251: any evidence as to whether the epipterygoid contributed to the basicranial recess (basipterygoid joint), as in Captorhinus, or not, as in Mesosaurus?

313: typo in ‘synsapids’

Gauthier, J. A. 1994. The diversification of the amniotes. Pages 129–159
in Major Features of Vertebrate Evolution (D. R. Prothero and R. M.
Schoch, eds.). Paleontological Society, Knoxville.

·

Basic reporting

There are problems both with the figures (the 3D reconstructions need to be smoothed out) and with the phylogenetic analysis (no multi-state characters were ordered, apparently, despite the fact that many are cline characters). See my comments in the section “General comments for the author ».

Experimental design

There are problems with the phylogenetic analysis: no multi-state characters were ordered, apparently, despite the fact that many are cline characters. See my comments in the section “General comments for the author ».

Validity of the findings

The problems of state ordering suggest that one of the main findings of the phylogenetic analysis is invalid; the results obtained by Laurin & Piñeiro (2018), which were ignored in the draft, do not support them. I suspect that this is partly because the authors don't like to publish a poorly-resolved consensus tree (I found that ordering the relevant characters, in this particular case, causes a loss of resolution). But certainly, this is preferable to publishing a better-resolved but erroneous tree or over-estimating support for their preferred hypothesis.See my comments in the section “General comments for the author ».

Additional comments

This draft redescribes an Early Permian parareptile through CT-scan data. It brings additional anatomical information compared to the original description and attempts at clarifying its phylogenetic position. It falls within the scope of PeerJ and in my opinion, should be published pending moderate revisions, which I summarise below. I have annotated the pdf file, so please ensure that the authors receive this attachment.

The description needs to be improved through an additional image processing step. The last step of the 3D image processing, the smoothing, was clearly not done, or not properly. This results in suboptimal images in which some texture appears, which looks like dermal sculpturing, but this is most likely purely artifactual. This is especially obvious in the lower jaws, with Macdougall & Reisz 2012 shows to have a fairly smooth ventral surface, wheras Figure 3 of the ms gives the impression of concentric grooved texture, like wood would have if differentially eroded. But this effect is visible on all anatomical figures. This should be corrected. For examples of how much better such images can look after smoothing, see, for instance Laloy et al. 2013 (though there is no need to cite the paper; I am just giving an example of what this should look like). And note that this last image processing step would actually highlight the true dermal sculpturing that is present on most elements, which is currently hidden by the false, artifactual texturing.

The phylogenetic part of the paper suffers from the omission of consideration of any of the detailed comments that Laurin & Piñeiro (2018) made on the matrix of MacDougall et al. (2018) that forms the basis of the analysis presented in the draft (the authors only modified the scores of a few cells of Feeserpeton). Most importantly, Laurin & Piñeiro (2018) pointed out that several characters should be ordered because they form morphoclines. Was this done here? This should be specified (it is not in the supplements either). This is mathematically obvious (though ignored by many systematists) and has been empirically demonstrated through simulations (Rineau et al., 2015, 2018). Both supplements and the paper itself suggest that all characters were left unordered, which is simply wrong (and not a matter of opinion, contrary to what many practicing systematists seem to believe). Yet, Laurin & Piñeiro (2018) demonstrated that ordering the characters reduced resolution, so the sister-group relationship between Lanthanosuchoidea and the large clade that includes Bolosauria, Procolophonoidea, Pareiasauridae, Nycteroleteridae, Nyctiphruretidae, and Microleter may not be supported by the data, if properly analyzed. That relationship was not supported by the strict consensus of the same data (except for the few changes in scores to Feeserpeton made in this draft) by the analysis made by Laurin & Piñeiro (2018), though it was weakly supported in the majority-rule consensus. If the authors carry out another analysis as I urge them to, I suggest that they also include the bootstrap percentages below 50% of the nodes that are in the tree; this is useful information.

On a more fundamental level, I find particularly troublesome to deliberately ignore colleague's comments because dialogue between scientists and the improvement in methods and accuracy of our data is at the core of the scientific procedure and a major (possibly most important) contributor to scientific progress. To deliberately ignore such comments is negligent at best, and dishonest at worst. I urge the authors to consider this carefully. The authors know about Laurin & Piñeiro (2018); Reisz (one of the co-authors) is the Chief Editor of the journal section in which it was published and we exchanged some e-mail messages about it before and during the review process.

Finally, note that I don't want to hide behind reviewer anonymity, which I consider a form of dishonesty and a hinderance to scientific dialogue. If the authors wish to discuss any of these issues (and the other, more minor issues raised in my annotations to both documents), they should feel free to contact me.

Best wishes,

Michel Laurin

References

Laloy F, Rage J-C, Evans SE, Boistel R, Lenoir N, and Laurin M. 2013. A re-interpretation of the Eocene anuran Thaumastosaurus based on microCT examination of a ‘mummified’ specimen. PLoS ONE 8:1–11. 10.1371/journal.pone.0074874
Laurin M, and Piñeiro G. 2018. Response: Commentary: A Reassessment of the Taxonomic Position of Mesosaurs, and a Surprising Phylogeny of Early Amniotes. Frontiers in Earth Science 6:220. 10.3389/feart.2018.00220
Macdougall MJ, and Reisz R. 2012. A new parareptile (Parareptilia, Lanthanosuchoidea) from the Early Permian of Oklahoma. J Vertebr Paleontol 32:1018–1026.
MacDougall MJ, Modesto SP, Brocklehurst N, Verrière A, Reisz RR, and Fröbisch J. 2018. Response: A Reassessment of the Taxonomic Position of Mesosaurs, and a Surprising Phylogeny of Early Amniotes. Frontiers in Earth Science 6:99. 10.3389/feart.2018.00099
Rineau V, Grand A, Zaragüeta R, and Laurin M. 2015. Experimental systematics: sensitivity of cladistic methods to polarization and character ordering schemes. Contributions to Zoology 84:129-148.
Rineau V, Zaragüeta I Bagils R, and Laurin M. 2018. Impact of errors on cladistic inference: simulation-based comparison between parsimony and three-taxon analysis. Contributions to Zoology 87:25-40.

·

Basic reporting

no comment

Experimental design

no comments

Validity of the findings

no comments

Additional comments

The manuscript is very clean, I try to look for errors or contradictions but only found some typos and minor omissions (see attached) that do not affect any the validity of the work.

---

## Round 0.2 · Minor Revisions

There are relatively minor recommendations for changes to the text of the MS including Figure 8's legend, and 1 reviewer still has reservations but does not feel they should hold up the MS further (however, their remarks deserve consideration and it would improve the MS to take those up). Please make these final changes and we should be able to accept the paper-- thank you!

·

Basic reporting

Good overall; some minor typos and a couple of minor omissions in references section (see ‘General comments for the author’).

Experimental design

no comment

Validity of the findings

no comment

Additional comments

As I communicated in my first review, this submission is a good, CT-based description of an early reptile and a re-analysis of its phylogenetic position, and it will be a very useful and well cited contribution to the paleontology of early amniotes.

I have a few comments and corrections:

line 79: the ‘d’ in ‘Macdougall’ should be capitalized; see also lines 193 and 484

line 82: I think ‘Middle’ should not be capitalized (it should be ‘middle Permian’); see also line 86

line 209: end of sentence should read ‘.. (Figs. 3, 4).’

line 225: typo in ‘intermanibularis’

line 310: I think the first comma should be a semi-colon

line 336: typo in ‘charcters’

line 339: typo in ‘pareisaurs’

line 368: this structure is more of a ‘coronoid eminence’ than a ‘coronoid process’

line 376: change ‘does not’ to ‘do not’ (or to: ‘the presence of multiple coronoids does not’)

line 428: add journal name, volume/issue, and page numbers

line 488: typo in ‘Unniversity’

line 508: typo in ‘Tazania’

line 515: the Zaher et al. paper was published in volume 5, pages 111-138

line 546: semi-colon should follow ‘supraoccipital’

·

Basic reporting

Now fine. But see my general comments to the authors.

Experimental design

Now fine.

Validity of the findings

Fine.

Additional comments

The revised draft has been sufficiently improved that I can recommend acceptance. The figures have been much improved by smoothing. From the rebuttal letter and the draft, I think that the authors are still a bit confused about ordering of multi-state characters because they state (in the letter) that do not wish to assume direction of evolution, but this is assumed only if you polarize a priori ; ordering assumes no direction . They also state (in the draft) that they “ did not want to make a priori assumptions regarding character evolution”, but assuming that any state can be transformed into any other is often a highly unrealistic assumption, which they apparently don't hesitate in making. However, given that they performed a second analysis with ordering, the basic information is provided (in the supplements), so points of view don't matter much. I only lament this misleading sentence (“we did not want to make a priori assumptions regarding character evolution”) because it amounts to saying that it is the most reasonable hypothesis to assume that a species could evolve from the size of a mouse directly to that of an elephant (without intermediate evolutionary steps) as easily as from the size of a mouse to that of a rat, but few proponents of fully unordered states realize this, and this paper conveys this misconception. However, it is so frequent in the literature, that this is like a drop in the ocean... Let it be.

In the page proofs (if the paper is accepted now, as I suggest), the authors might want to clarify, at the beginning of the phylogenetic analysis section (starting at line 127), that they performed two analyses because this is unannounced till line 148. Also, in the legend of figure 8, they should indicate that this is the result of the first phylogenetic analysis (all states unordered). As it stands, readers will have to look at the supplement to understand what Figure 8 represents because its legend indicates « from the phylogenetic analysis ». But there are two such analyses, so the legend is ambiguous. It is interesting to see that with ordered states, the resolution actually improved, contrary to their earlier matrix ; of course, ordering states has an unpredictable effect on resolution, compared to not ordering them (it can either increase or decrease it).

Best wishes,
Michel Laurin

---

## Round 0.3 · accepted · Accept

Thank you for your revised MS, which now satisfies the reviewers' comments and is acceptable--- congrats!